# wMel *Wolbachia* alters female post-mating behaviors and physiology in the dengue vector mosquito *Aedes aegypti*

Jessica Osorio[1], Sara Villa-Arias[1,2], Carolina Camargo[3], Luis Felipe Ramírez-Sánchez [1], Luisa María Barrientos[1], Carolina Bedoya[1], Guillermo Rúa-Uribe[4], Steve Dorus[5], Catalina Alfonso-Parra [1,2✉] & Frank W. Avila [1✉]

Globally invasive *Aedes aegypti* disseminate numerous arboviruses that impact human health. One promising method to control *Ae. aegypti* populations is transinfection with *Wolbachia pipientis*, which naturally infects ~40–52% of insects but not *Ae. aegypti*. Transinfection of *Ae. aegypti* with the wMel *Wolbachia* strain induces cytoplasmic incompatibility (CI), allows infected individuals to invade native populations, and inhibits transmission of medically relevant arboviruses by females. Female insects undergo post-mating physiological and behavioral changes—referred to as the female post-mating response (PMR)—required for optimal fertility. PMRs are typically elicited by male seminal fluid proteins (SFPs) transferred with sperm during mating but can be modified by other factors, including microbiome composition. *Wolbachia* has modest effects on *Ae. aegypti* fertility, but its influence on other PMRs is unknown. Here, we show that *Wolbachia* influences female fecundity, fertility, and re-mating incidence and significantly extends the longevity of virgin females. Using proteomic methods to examine the seminal proteome of infected males, we found that *Wolbachia* moderately affects SFP composition. However, we identified 125 paternally transferred *Wolbachia* proteins, but the CI factor proteins (Cifs) were not among them. Our findings indicate that *Wolbachia* infection of *Ae. aegypti* alters female PMRs, potentially influencing control programs that utilize *Wolbachia*-infected individuals.

[1] Max Planck Tandem Group in Mosquito Reproductive Biology, Universidad de Antioquia, Medellín, Colombia. [2] Instituto Colombiano de Medicina Tropical, Universidad CES, Sabaneta, Colombia. [3] Centro de Investigación de la caña de azúcar CENICAÑA, Valle del Cauca, Colombia. [4] Grupo Entomología Médica, Universidad de Antioquia, Medellín, Colombia. [5] Center for Reproductive Evolution, Syracuse University, Syracuse, USA. ✉email: catalfonso@gmail.com; grupotandem.mosquito@udea.edu.co

*A*edes aegypti mosquitoes are a globally invasive species that have successfully colonized large portions of the tropics and subtropics[1,2]. *Aedes aegypti* has a propensity for colonizing urban environments[3,4], and females of this species have a preference for human hosts[5,6], factors that have facilitated the transmission of viruses spread by this species, which include the dengue[7], Zika[8], chikungunya[9] and yellow fever viruses[10]. The habitable territory of *Ae. aegypti* is predicted to expand with rising global temperatures[2,11] and increased urbanization[6,12], making control of this species essential to mitigate its impact on human health.

Efforts to control *Ae. aegypti* have historically relied on insecticide use. However, the increased insecticide resistance of *Ae. aegypti* populations have reduced the efficacy of chemical control[13], necessitating the development of novel control methods. One promising method is the transinfection of *Ae. aegypti* with the obligate intracellular bacterium *Wolbachia pipientis*, a symbiont that naturally infects 40–52% of insect species[14,15], but not *Ae. aegypti*. *Wolbachia* is maternally inherited and induces cytoplasmic incompatibility (CI) in transinfected *Ae. aegypti*[16,17], a phenomenon where uninfected females that mate with infected males do not produce viable progeny, while infected females produce viable, *Wolbachia*-infected progeny regardless of the infection status of their mates. The induction of CI allows infected *Ae. aegypti* to rapidly spread into uninfected populations[16–18], where they remain stable long-term[19]. *Wolbachia* infection also suppresses arbovirus transmission by *Ae. aegypti* females, including DENV, ZIKV and CHIKV[17,20–29].

Given that *Wolbachia*-infected *Ae. aegypti* are able to quickly disseminate into mosquito populations, *Wolbachia*-infected populations—compromised for their ability to transmit disease—can effectively replace native populations upon the release of infected males and females into the environment[30]. Alternatively, the continuous release of *Wolbachia*-infected males can reduce native mosquito populations through the establishment of CI[31,32]. Population replacement programs utilizing *Ae. aegypti* infected with the wMel *Wolbachia* strain isolated from *Drosophila melanogaster* have been initiated in several areas where DENV transmission occurs[33–36], including in Medellín, Colombia[37]. The successful establishment of infected populations is dependent on a minimal effect of *Wolbachia* on the reproductive parameters of liberated *Ae. aegypti* adults. wMel *Wolbachia* has been reported to have no effects[38] or modest effects on *Ae. aegypti* fertility[39], but how *Wolbachia* might alter other reproductive processes in *Ae. aegypti* has not been explored.

Mating induces physiological and behavioral changes in female insects that facilitate the production of progeny[40,41]. Female *Ae. aegypti* post-mating responses (PMRs) include an inhibition to re-mating[42], increased female longevity[43], increased oviposition rates[43], and changes in gene expression in female reproductive tissues[44–46]. The primary effectors of the *Ae. aegypti* female PMR are seminal fluid proteins (SFPs)[43,47,48] transferred to the female reproductive tract along with sperm during mating. Female PMRs in insects are also influenced by other factors, such as male age[49], adult nutrition[50], and adult microbiome composition[51,52]. wMel *Wolbachia* has been shown to alter female PMRs in *D. melanogaster*[53], which may be due to the observed modification of SFP composition in infected males[54]. However, *Wolbachia* infection alters protein secretion from the *D. melanogaster* female sperm storage organs[54], which express genes essential for ovulation, oviposition, sperm motility, and sperm storage[55,56]. Thus, *Wolbachia* infection could potentially influence female *Ae. aegypti* post-mating changes in a sex-specific manner.

In the present study, we examined how wMel *Wolbachia* influences *Ae. aegypti* female PMRs. We collected wMel *Wolbachia*-infected adults being released in Medellín, Colombia[37] and backcrossed infected females to our laboratory strain for seven generations to generate a *Wolbachia*-infected colony in the same genetic background. We examined how *Wolbachia* influences fecundity, fertility, re-mating incidence, and female longevity. Although *Wolbachia* infection had no effect on sperm transfer during mating or the storage of sperm by mated *Ae. aegypti* females, fecundity, fertility, and re-mating incidence were impacted. Additionally, female longevity was altered in *Wolbachia*-positive *Ae. aegypti* females independent of mating. We used proteomic methods to examine SFP levels transferred to females by *Wolbachia*-infected males, finding that *Wolbachia* has a modest effect on SFP composition. Our proteomic analysis also allowed us to identify *Wolbachia* proteins paternally transferred to females during mating. To our surprise, although 125 *Wolbachia* proteins were identified, our analysis did not reveal the presence of the CI factor (Cif) proteins that modify sperm to establish CI[57]. Our results show that the presence of *Wolbachia* in *Ae. aegypti* alters adult fertility and influences female post-mating behaviors and physiology. The effects we report here have potential implications for population replacement programs[30,58] and population suppression programs[31,32] that use *Wolbachia*-infected *Ae. aegypti* to control mosquito populations or suppress disease transmission.

## Results

**Wolbachia impacts fecundity and fertility of female *Aedes aegypti*.** To examine the effects of *Wolbachia* infection on the *Ae. aegypti* female PMR, we collected wMel *Wolbachia*-infected individuals from the field[37] and backcrossed them to Thai strain[59] *Ae. aegypti*, generating the Thai^Wolb strain. We examined the potential impact of *Wolbachia* on the fecundity and fertility of female *Ae. aegypti*, as wMel *Wolbachia* has moderate sex-specific effects on fertility in this species[38,39]. Given that female size influences fecundity[60], we first examined the size of the Thai and Thai^Wolb adults used in our assays, finding that Thai and Thai^Wolb adults were similarly sized when reared under the same conditions (Supplementary Fig. 1). To assess possible male- or female-specific effects of *Wolbachia* infection on parameters of fertility, we performed our assays in all mating combinations (shown as female x male): Thai x Thai (control), Thai x Thai^Wolb, Thai^Wolb x Thai, Thai^Wolb x Thai^Wolb.

We found significant differences in fecundity between the different mating combinations (DF = 3, F = 17.78, $p < 0.0001$; Fig. 1a). We did not detect an effect of male infection on female fecundity, as Thai females laid a similar quantity of eggs when mated to Thai or Thai^Wolb males ($p = 0.95$; Fig. 1a). However, Thai^Wolb females laid significantly fewer eggs than Thai females when mated to Thai males ($p = 0.007$; Fig. 1a) and suffered a further reduction in fecundity upon mating to Thai^Wolb males ($p < 0.0001$; Fig. 1a). Similarly, fertility (shown as hatch percentage) was significantly different between all mating combinations (DF = 3, F = 146.19, $p < 2.2e\text{-}16$; Fig. 1b). The fertility of Thai^Wolb females mated to Thai males was significantly reduced compared to control matings ($p < 0.0001$; Fig. 1b), with the largest suppression of fertility observed when both sexes were infected ($p < 0.0001$; Fig. 1b). As expected, we observed a significant reduction in fertility when Thai females mated to Thai^Wolb males due to the establishment of CI ($p < 0.0001$; Fig. 1b).

**Female re-mating incidence increases after mating with *Wolbachia*-infected males.** We next examined whether *Wolbachia* infection affects a male's ability to inhibit re-mating by their mates. In our assays, 27% of Thai females re-mated when initially mated to Thai males (Table 1), similar to previous reports using this strain[42,49]. When mated to Thai^Wolb males, however, we

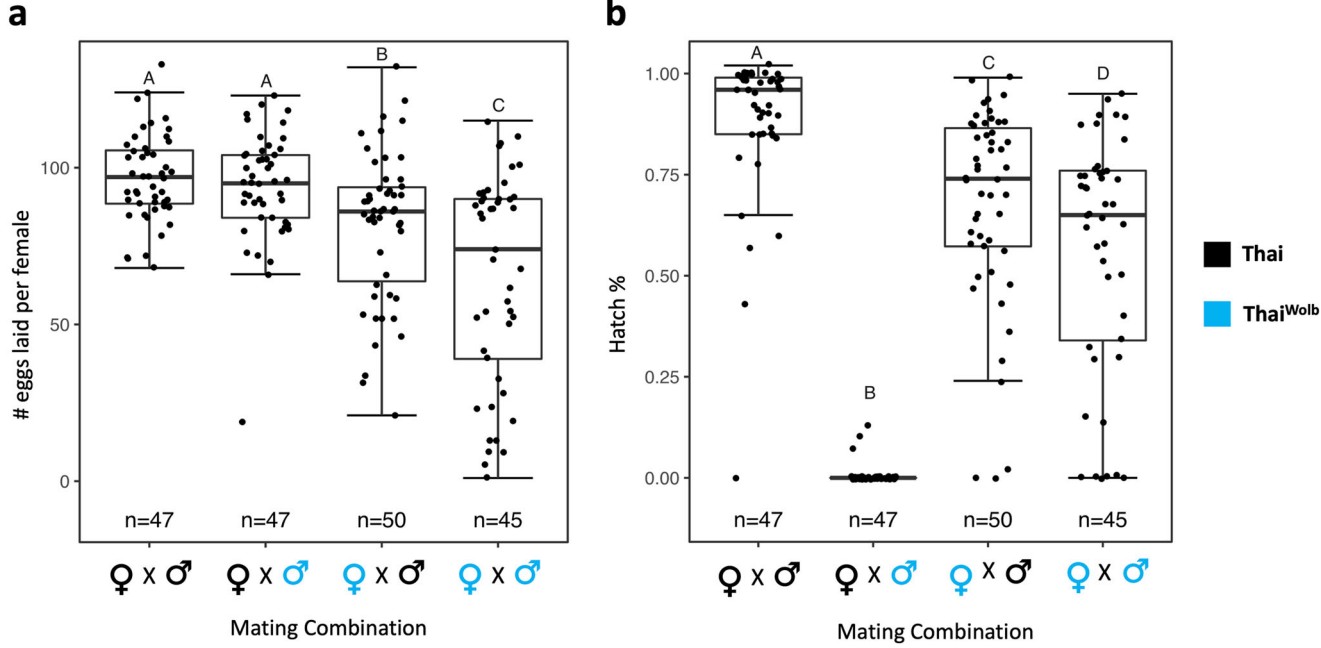

**Fig. 1 *Wolbachia* impacts fecundity and fertility of female *Aedes aegypti*.** Fecundity (**a**) and hatch percentage (**b**) for each mating combination. Groups denoted by different letters are significantly different for a post hoc Tukey test ($p < 0.05$). For the box plots, the middle horizontal line represents the median, the lower and upper margins of the box represent the 25th and 75th quartiles, and the whiskers extend to the minimum and maximum of the data (excluding outliers, shown as points outside the whiskers).

**Table 1 Female re-mating incidence after mating to a *Wolbachia*-infected male.**

| Female | 1st mating male | N | % re-mated |
|---|---|---|---|
| Thai | Thai | 194 | 27.3% |
| | Thai[Wolb] | 189 | 37.6% |
| Thai[Wolb] | Thai | 192 | 30.7% |
| | Thai[Wolb] | 197 | 36.4% |
| Acacias | Thai | 96 | 35.4% |
| | Thai[Wolb] | 96 | 43.8% |
| Rockefeller | Thai | 98 | 40.6% |
| | Thai[Wolb] | 96 | 45.8% |

Re-mating incidence of the indicated female strain after an initial mating to a Thai or Thai[Wolb] male.

observed a significant increase in re-mating incidence ($\chi^2 = 4.5$, DF = 1, $p = 0.03$; Table 1). Thai[Wolb] females also re-mated at significantly higher rates when first mated to a Thai[Wolb] male compared to an initial mating with a Thai male ($\chi^2 = 3.2$, DF = 1, $p = 0.04$; Table 1). Given the increase in re-mating incidence observed after initially mating with Thai[Wolb] males, we next evaluated if this effect is detectable upon mating with females of a different strain. We mated Thai and Thai[Wolb] males to *Ae. aegypti* collected in Acacias, Colombia[61] and to Rockefeller strain females. In both strains, we observed a similar trend: females initially mated to Thai[Wolb] males re-mated at higher rates than those initially mated to Thai males, although in each case, the effect was not significant (Acacias: $\chi^2 = 1.3$, DF = 1, $p = 0.3$; Rockefeller: $\chi^2 = 0.5$, DF = 1, $p = 0.5$; Table 1). Although Thai[Wolb] males were less able to prevent re-mating by Thai females shortly after an initial mating, they induced complete refractoriness in their mates by 24 h, and refractoriness was maintained for 7 days after an initial mating (Supplementary Table 1) when a significant proportion of Thai females re-mate if they receive insufficient SFP quantities during mating[60].

Multiply mated Thai strain *Ae. aegypti* females produce mixed progeny by utilizing sperm from the first and second mating males, although they display first male precedence[49]. This suggests that females initially mated to a *Wolbachia*-infected male will generate viable progeny upon re-insemination by a second, uninfected male. We hatched eggs laid by re-inseminated females in our assays to determine if they produced viable progeny. Thai, Acacias and Rockefeller females initially mated to a Thai[Wolb] male each generated viable progeny if they subsequently re-mated with an uninfected male, although female fertility was significantly reduced compared to females initially mated to a Thai male (Supplementary Fig. 2).

**Wolbachia extends the lifespan of virgin *Aedes aegypti* females.** As wMel *Wolbachia* increases female lifespan in *D. melanogaster*[62] and in wMel transinfected male *Ae. albopictus*[63], we asked if *Wolbachia* infection alters *Ae. aegypti* lifespan. Mated *Ae. aegypti* females have significantly longer lifespans than virgins[43,64], an effect of SFP receipt[43]. We examined matings between infected or uninfected individuals, observing that mated Thai females lived significantly longer than virgins ($p = 4e-08$; Fig. 2a) as previously reported[43]. However, the lifespan of virgin and mated Thai[Wolb] females did not significantly differ ($p = 0.2$; Fig. 2a), and the longevity of virgin Thai[Wolb] females was significantly greater than virgin Thai females ($p = 2e-12$; Fig. 2a). The observed increase in longevity was specific to females, as *Wolbachia* infection had no effect on male lifespan ($p = 0.86$; Supplementary Fig. 3). As host nutrition can alter the effects of *Wolbachia* on the lifespan of the host[65], we asked if adult nutrition affected longevity of Thai[Wolb] virgin females. We examined the longevity of virgin females with access to 10% or 2% sucrose. We found that Thai ($p = 9.895e-07$) and Thai[Wolb] virgin females ($p = 5.065e-08$) lived significantly longer on 10% sucrose compared to 2% sucrose (Fig. 2b). However, on each diet, Thai[Wolb] virgin females lived significantly longer than Thai

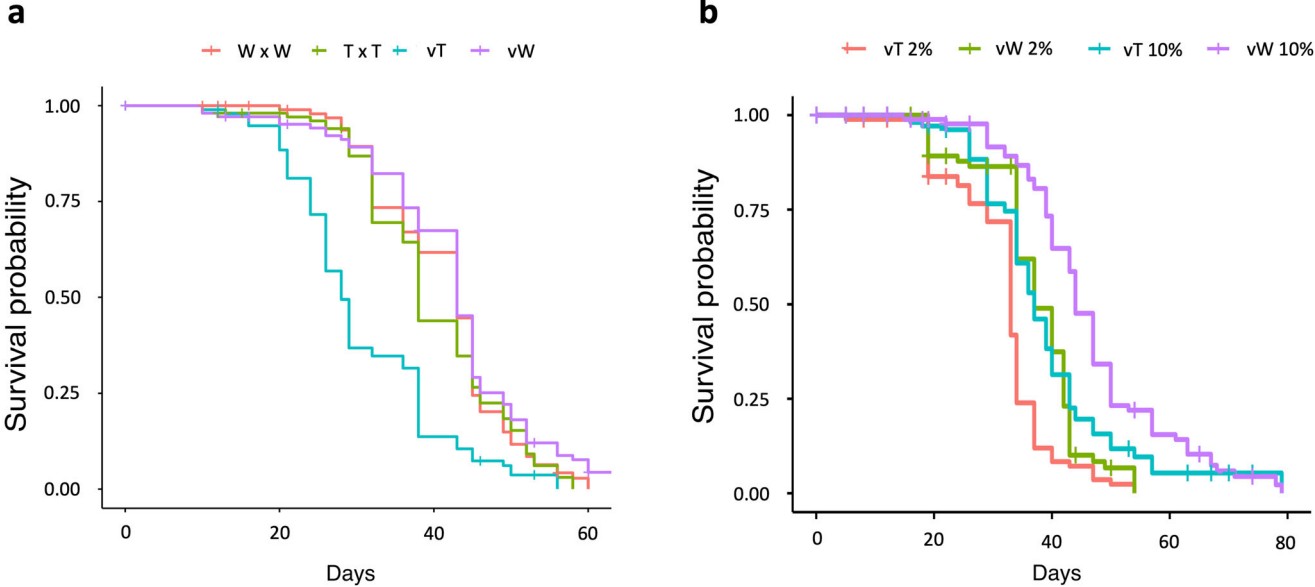

**Fig. 2 *Wolbachia* increases longevity of virgin females. a** Longevity of virgin and females mated to males from the same colony ($N_{T \times T} = 98$, $N_{W \times W} = 93$, $N_{vT} = 94$, $N_{vW} = 100$). **b** Longevity of virgin females with access to 10% or 2% sugar as adults ($N_{vT2\%} = 84$, $N_{vT10} = 97$, $N_{vW2} = 68$, $N_{vW10} = 79$). T = Thai, W = Thai^Wolb, vT = virgin Thai, vW = virgin Thai^Wolb.

females (10% sucrose: $p = 0.0003$; 2% sucrose: $p = 1.64e{-}05$; Fig. 2b).

**Wolbachia does not influence sperm quantity transferred by males, or sperm quantity stored by females.** *Wolbachia* impacts sperm production[66] and sperm quantity transferred[67] by *D. simulans* males, raising the possibility that *Wolbachia* infection impacts sperm production and/or sperm transfer by Thai^Wolb males, which would impact the quantity of sperm stored by females[68]. However, we did not detect differences in the quantity of sperm present in the male seminal vesicles—the organs that store mature sperm in the male reproductive tract[69]—of Thai^Wolb males compared to Thai males (DF = 1, F = 0.752, $p = 0.39$; Fig. 3a), similar to results for *Ae. aegypti* infected with the pathogenic *Wolbachia* strain wMelPop[39]. The quantity of sperm transferred by Thai^Wolb males during mating also did not differ from uninfected Thai males (DF = 3, F = 0.418, $p = 0.741$; Fig. 3b). Finally, we assessed the quantity of sperm females stored in their spermathecae, the long-term sperm storage organs[70]. Sperm quantity in the spermathecae was similar in the spermathecae of Thai and Thai^Wolb females 24 h after insemination (DF = 3, F = 2.039, $p = 0.114$; Fig. 3c). Thus, *Wolbachia* infection does not appear to influence sperm production, sperm transfer during mating, or female sperm storage.

**Wolbachia-dependent changes in the composition of SFPs transferred at mating.** wMel *Wolbachia* alters the expression of genes that code for SFPs[53] and changes SFP composition in naturally infected *D. melanogaster* males[54]. Given that Thai^Wolb males are suboptimal in preventing subsequent copulations by their mates (Table 1), and re-mating inhibition is mediated by SFP receipt[48], we asked if SFP composition in *Wolbachia*-infected males differs from uninfected males. To identify seminal proteins, we used a proteomic approach that allows for the identification of male proteins transferred to females during mating[71,72]. Females labeled with the natural isotope ¹⁵N were mated to unlabeled, normally reared Thai and Thai^Wolb males, and seminal proteins isolated from the bursae immediately after insemination were identified by tandem mass spectrometry (LC-MS/MS) and their

abundances quantified (see "Methods"). To differentiate between sperm proteins and SFPs, we used the *Ae. aegypti* sperm and SFP proteome reported in ref. [72].

We found that *Wolbachia* infection had a modest effect on the composition of seminal proteins transferred during mating, as Thai^Wolb and Thai male ejaculates had similar abundances of SFPs, sperm proteins, and sperm/SFP overlapping proteins (i.e., proteins identified in both the sperm and SFP proteomes, but more abundant in the ejaculate relative to sperm[72]) (Fig. 4a–d). However, one seminal protein was significantly more abundant in Thai males compared to Thai^wolb males: the SFP trypsin-7 (AAEL006429) (Fig. 4b). Further, although not statistically significant, three additional seminal proteins were found to be more than fourfold more abundant in Thai male ejaculates compared to Thai^Wolb males, the sperm protein AAEL024468 (Fig. 4c), and two unclassified proteins, AAEL006115 and AAEL003365 (Fig. 4a); both AAEL006115 and AAEL003365 are co-expressed in the *Ae. aegypti* testes and male accessory gland[72]. Finally, we identified seminal proteins that were only detected in Thai or Thai^Wolb male ejaculates but not detected in the other (Supplementary Table 2).

**Identification of paternally transferred *Wolbachia* proteins.** Our proteomic labeling method also allowed us to identify *Wolbachia* proteins present in the seminal fluid and transferred to females during mating. We identified 125 wMel *Wolbachia* proteins across all replicates that were present in Thai^Wolb male ejaculates (Supplementary Data 1). However, only 20 of these proteins were consistently detected in all replicates (Table 2). Paternally transferred wMel *Wolbachia* proteins fell into several functional categories (Supplementary Data 1), and Gene Ontology (GO) analysis showed enrichment for ATP- and nucleotide-binding proteins, P-loop containing triphosphate hydrolases, and translocases (Supplementary Fig. 4). Our analysis did not identify Cif proteins that recapitulate CI in a transgenic system[73]. However, we identified proteins that can modify the CI phenotype[74] (Supplementary Table 3), including WD0462, a predicted *Wolbachia* effector molecule[75]. Several paternally transferred *Wolbachia* proteins are derived from phage WO sequences integrated

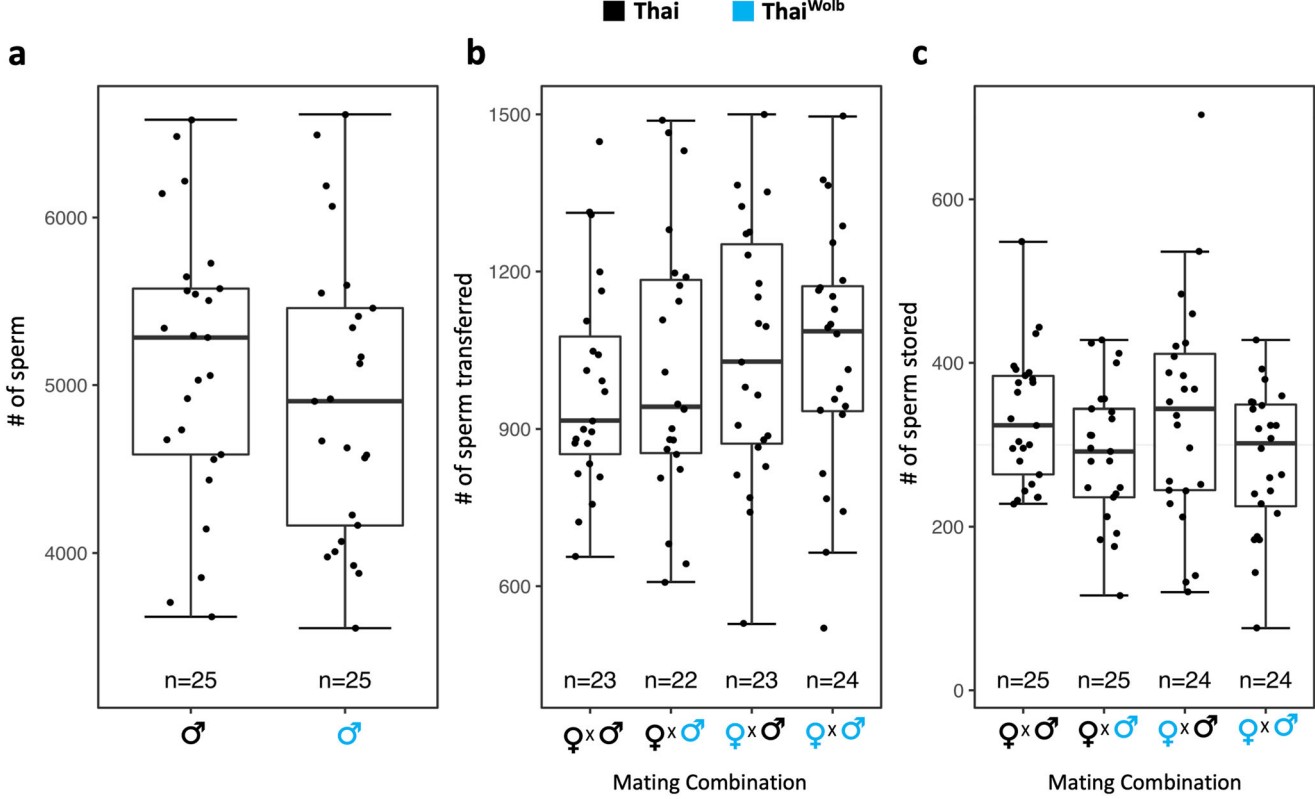

**Fig. 3 *Wolbachia* does not impact sperm production, sperm transfer, or female sperm storage.** Sperm quantities in the seminal vesicle of virgin males (**a**) transferred to the bursa of the female reproductive tract during mating (**b**) and stored in the spermathecae of mated females (**c**). For the box plots, the middle horizontal line represents the median, the lower and upper margins of the box represent the 25th and 75th quartiles, and the whiskers extend to the minimum and maximum of the data (excluding outliers, shown as points outside the whiskers).

into the *Wolbachia* genome or WO-like Islands that are associated with and/or derived from phage WO[76] (Supplementary Table 3). Further, we detected transferred phage WO proteins that are associated with the Eukaryotic Association Module (Supplementary Table 3), a region of the phage genome that contains genes that code for proteins with eukaryotic domains and are predicted to interact with the host[77].

## Discussion

*Wolbachia* is a common insect symbiont that can modify host physiology and behaviors. Given that *Wolbachia* alters reproductive outcomes that favor its propagation, it is being used as a tool to reduce the vector competency of *Ae. aegypti* females[58] and offers an alternative to continued insecticide use and/or the release of genetically modified mosquitoes. Control programs are currently conducting large-scale releases of *Wolbachia*-infected males to suppress vector populations[31,32] or both sexes in population replacement programs[30,58]. The success of these programs depends on *Wolbachia* having minimal effects on infected individuals to allow liberated *Wolbachia*-infected males to successfully mate with native females or for liberated *Wolbachia*-infected adults to invade populations targeted for replacement. While studies have assessed the effects of wMel *Wolbachia* infection on *Ae. aegypti* fertility[26,38,39], no studies have reported how *Wolbachia* influences other female post-mating responses in this important disease vector.

The effects of wMel *Wolbachia* infection on some female PMRs have been documented in *Drosophila*, but the impact of *Wolbachia* on the female PMR in transinfected species is only beginning to be dissected. We found that wMel *Wolbachia* altered *Ae.*

*aegypti* female PMRs similar to naturally infected *Drosophila* and transinfected insects in some regards but differed in others (Table 3). The genetic background of the *Drosophila*[78] and rearing conditions[62] each influence the effects of *Wolbachia* infection on the host, suggesting that similar effects will be observed in transinfected individuals. wMel transinfected *Ae. aegypti* display a suppression in fertility when both sexes are infected but have also noted effects when only one sex is infected (Table 3). We did not observe an effect of male infection but found that *Wolbachia* lowers the fecundity and fertility of *Ae. aegypti* females, regardless of the infection status of their mates and is further suppressed after mating with an infected male. Mating[44–46], SFPs[47] and blood-feeding[45] each modify gene expression in the *Ae. aegypti* female reproductive tract, including genes expressed from the female sperm storage organs[45,46] whose products are essential for fertility[46,55,79]. Although *Wolbachia* alters protein production from the sperm storage organs in *Drosophila*[54], it is unknown if a similar effect occurs in *Ae. aegypti*, which may account for the observed reduction in fertility.

wMel *Wolbachia* had a mating-independent effect on *Ae. aegypti* female, but not male, lifespan. The post-mating increase in female longevity was absent in wMel-infected females, but the longevity of virgin Thai^Wolb females was significantly increased. *Wolbachia* increases longevity in other insects, including wMel transinfected male *Ae. albopictus*[63] and *D. melanogaster*, the latter dependent on the *Drosophila* strain assessed[78]. Fitness costs or benefits of *Wolbachia* infection often differ between studies, possibly due to differences in rearing conditions or host genetic background. The reasons for the increase in *Aedes* longevity are unclear. Insulin signaling is associated with lifespan in a number of organisms[80], including *D. melanogaster*, where increased

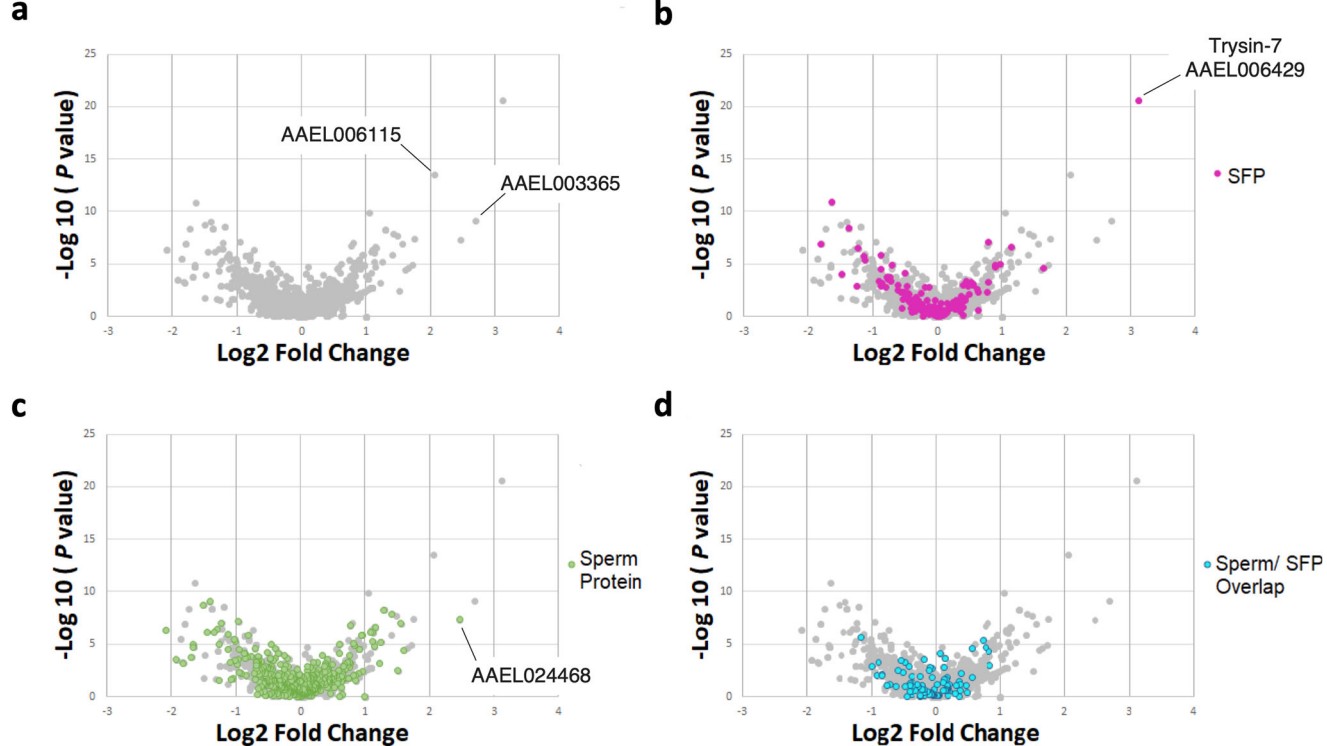

**Fig. 4 Composition of seminal proteins transferred by Thai^Wolb and Thai males.** Volcano plots showing the abundance of all proteins identified in Thai and Thai^Wolb ejaculates (**a**), seminal proteins classified as SFPs (**b**), sperm proteins (**c**), and sperm/SFP overlapping proteins (**d**). Values below zero represent proteins with higher abundance in Thai^Wolb male ejaculates, and values above zero represent proteins with higher abundance in Thai male ejaculates.

| Table 2 Paternally transferred *Wolbachia* proteins identified in all biological replicates. | | |
|---|---|---|
| **Accession** | **Gene** | **Protein** |
| _ATPB_WOLPM | atpD | Synthase subunit betaATP synthase subunit beta |
| _DNAK_WOLPM | dnaK | Chaperone protein DnaK |
| Q73H52_WOLPM | WD0722 | Ammonium transporter |
| NUOD_WOLPM | nuoD | NADH-quinone oxidoreductase subunit D |
| Q73HV3_WOLPM | sdhA | Succinate dehydrogenase flavoprotein subunit |
| Q73GH8_WOLPM | WD0976 | NADH-quinone oxidoreductase subunit F |
| ATPA_WOLPM | atpA | ATP synthase subunit alpha |
| RPOBC_WOLPM | rpoBC | Bifunctional DNA-directed RNA polymerase subunit beta-beta |
| HTPG_WOLPM | htpG | Chaperone protein HtpG |
| Q73GV1_WOLPM | cutA | Periplasmic divalent cation tolerance protein |
| EFTU2_WOLPM | tuf2 | Elongation factor Tu 2 |
| EFTU1_WOLPM | tuf1 | Elongation factor Tu 1 |
| Q73FW7_WOLPM | WD1206 | Biotin transporter |
| Q73IK0_WOLPM | nuoG | NADH-quinone oxidoreductase |
| Q73HL2_WOLPM | sucB | Dihydrolipoyllysine-residue succinyltransferase component of 2-oxoglutarate dehydrogenase complex |
| Q73HQ4_WOLPM | fumC | Fumarate hydratase class II |
| Q73GU2_WOLPM | WD0838 | Glycerophosphoryl diester phosphodiesterase |
| Q73FY5_WOLPM | WD1187 | Conserved domain protein |
| Q73HL3_WOLPM | hemC | Porphobilinogen deaminase |
| NUOB_WOLPM | nuoB | NADH-quinone oxidoreductase subunit B |

insulin signaling reduces mated female lifespan[81]. In *Ae. aegypti*, insulin-like peptides (ILPs) have been implicated in altering female lifespan, with a reduction in ILPs increasing female longevity[82]. Further analysis is required to determine how *Wolbachia* might interact with insulin signaling or other pathways to modulate female lifespan in *Ae. aegypti*.

We also observed that *Wolbachia*-infected males were less successful than their non-infected counterparts at inhibiting re-mating by their mates. The suppression of re-mating is a key *Ae.*

*aegypti* female PMR. The probability of re-mating is highest within the first 2 h of an initial mating[42], but declines with increased time—females are completely refractory by ~20 h after an initial mating[42] and do not mate again after this time[42,48]. Our assays may underestimate re-mating incidence, as a recent study using microsatellite markers to assign parentage found that *Ae. aegypti* females mate with up to four partners[83]. Given that SFP receipt mediates female re-mating incidence[48] and that wMel *Wolbachia* changes SFP composition in naturally infected *D.*

**Table 3 The effects of wMel *Wolbachia* on female post-mating responses in naturally infected *D. melanogaster* and transinfected insects.**

| Female PMR | Insect | Observation |
|---|---|---|
| **Fecundity** | *D. melanogaster* | Significantly increased[78,106] |
| | | Temperature dependent[62] |
| | | Female age dependent[62] |
| | | Male-specific effect on the first day of egg-laying[53] |
| | | No effect[107] |
| | *Ae. aegypti* | Significantly decreased[26,39,currentstudy] |
| | | Female-specific effect[39,currentstudy] |
| | | No effect[17,38,108] |
| | *Ae. albopictus* | Significantly increased[63] |
| | *Haematobia irritans exigua* | Significantly decreased[109] |
| | *D. nigrosparsa* | No effect[110] |
| **Fertility**[a] | *Ae. aegypti* | Significantly decreased[39,currentstudy] |
| | | Female-specific effect[39,currentstudy] |
| | | No effect[17,108] |
| | *Bemisia tabaci* | No effect[111] |
| **Longevity** | *D. melanogaster* | Significant increase in female longevity[106] |
| | | Temperature and sex dependent[62] |
| | | Strain dependent[78] |
| | | Male effect on female longevity[53] |
| | *Ae. aegypti* | Increased virgin female longevity[current study] |
| | | Decreased mated female longevity[17] |
| | | Strain dependent[108] |
| | *Ae. albopictus* | Increased male longevity[63] |
| **Re-mating incidence** | *D. melanogaster* | Male-specific effect on female re-mating incidence[53] |
| | *Ae. aegypti* | Male-specific effect on female re-mating incidence[current study] |

Reported results are derived from matings between infected individuals unless noted as a male- or female-specific effect.
[a]Ref. [111] used progeny counts to assess fertility, while refs. [17,39,108] and the current study used the percentage of laid eggs that hatched.

*melanogaster* males[54], we hypothesized that *Wolbachia* alters SFP composition in *Ae. aegypti*. SFP quantification detected moderate changes in SFP composition in this species. However, our analysis may not have identified proteolytically cleaved SFPs. Proteolysis of SFPs is common[84] and is often required for, or enhances, SFP function[85–87]. Cleavage of SFPs occurs in transit to, or quickly after deposition into the female reproductive tract[87]. The identification of proteolytically cleaved SFPs using bioinformatic methods is difficult without knowledge of the resulting cleavage products. Additionally, post-translational modification of SFPs, which can be necessary for their proper function[88], may be abnormal in *Wolbachia*-infected males. Further exploration is required to determine why *Wolbachia*-infected males are less able to prevent re-mating in their mates.

*Wolbachia* infection did not alter the quantity of sperm detected in the male reproductive tract or the quantity of sperm transferred during mating, suggesting that sperm production is unperturbed by *Wolbachia* in the testes. One aspect we did not assess is whether *Wolbachia* might affect sperm quality, as sperm function may be impacted by modifications made by *Wolbachia* Cif proteins during spermatogenesis[57] or be affected by a potential increase in reactive oxygen species that occurs in the testes of *Wolbachia*-infected *Drosophila*[89,90]. Sperm competitive ability is reduced in *Wolbachia*-infected *D. simulans* males[91], suggesting that an intrinsic property of sperm may be affected by

*Wolbachia*. Properties such as length and swim velocity influence sperm competitive outcomes in multiply-mated females[92]. The competitive ability of sperm from *Wolbachia*-infected *Ae. aegypti* males needs to be further examined, which may identify subtle defects in sperm ability not detected in our assays.

Our proteomic experiment identified 125 *Wolbachia* proteins that are paternally transferred during mating. Although the wMel proteins responsible for CI establishment are known[57,73], the molecular mechanism for the phenomenon has not been fully elucidated. Two models have been proposed for the establishment of CI: host modification and toxin antidote. The host-modification model suggests that Cif proteins modify sperm, modifications that are rescued by infected females. The toxin-antidote model suggests that Cifs are transported to the female via sperm but are inhibited by a rescue factor present in infected females that bind the Cifs and inhibit their toxicity[93]. We did not detect Cif proteins in the ejaculates of *Wolbachia*-infected males, providing support for the host-modification model. However, we cannot rule out that Cif protein abundance might be low in artificially infected *Ae. aegypti*, have undergone modification and/ or started to degrade, thereby limiting our detection abilities. CifA and CifB have been detected in mature spermatozoa of wMel-infected *D. melanogaster*[57], while CidB has been detected in mature spermatozoa of *Culex* males naturally infected with wPip[94]. Given that *Ae. aegypti* are artificially infected with *Wolbachia*, Cif proteins might not display the same properties observed in naturally infected insects. It would be interesting to examine wMel CifA and CifB localization patterns in developing and mature *Ae. aegypti* sperm to determine if they behave similarly to that what has been reported in naturally infected *D. melanogaster*[57].

For the success of programs that utilize *Wolbachia*-infected individuals to suppress or replace native *Ae. aegypti* populations, it is essential to understand how *Wolbachia* interacts with the reproductive processes of *Ae. aegypti*, including the induction of female PMRs. Our results show that *Wolbachia* alters some female PMRs, with the decrease in male ability to prevent further copulations potentially complicating the efficiency of population suppression programs or the successful establishment of liberated adults where population replacement is attempted. Continued investigation is necessary to determine whether the effects of wMel *Wolbachia* on female PMRs are also observed in *Ae. aegypti* transinfected with other *Wolbachia* strains used in control efforts[95], and to determine the molecular pathways impacted by *Wolbachia* infection to modify post-mating behaviors and physiology in female *Ae. aegypti*.

## Methods

**Mosquitoes**. Thai[59], DsRed[96], Acacias[61], and Rockefeller[61] strain *Ae. aegypti* were used in our assays. The Acacias strain of *Ae. aegypti* has been previously shown to be highly resistant to pyrethroid insecticides, while the Rockefeller strain is highly susceptible[61]. DsRed mosquitoes contain a transgene that labels sperm with the red fluorescent protein DsRed (Aaβ2t::DsRed)[96]. Mosquito eggs were hatched under a vacuum (−50 kPa) for 30 min. Larvae were reared at a density of 200/L in type II $H_2O$ supplemented with four (7.2–8.2 mm) Hikari Gold Cichlid food pellets (Hikari, Himeji, Japan). This feeding regimen produces adults of similar size[49,60]. $^{15}N$-labeled females were reared with a yeast slurry (see below). Pupae were transferred to 5 mL tubes to ensure virginity, and resulting adults were separated into sex-specific cages upon eclosion. Larvae and adults were kept in incubators at 27 °C, 70% relative humidity and a 12:12 h photoperiod. Adults had access to 10% sucrose ad libitum. Four- to six-day-old adults were used in our assays except for $^{15}N$-labeled females, which were mated at 2 days old. Wing lengths were measured as in ref. [97] to estimate individual size; wing lengths of the mosquito strains used in our assays are shown in Supplementary Fig. 1.

**Generation of *Wolbachia*-infected Thai strain *Aedes aegypti***. We collected *Ae. aegypti* infected with the wMel strain of *Wolbachia* being released in Medellín, Colombia[37] (we obtained permission to collect field specimens from the Secretaria de Salud de Medellín). Ovitraps[98] were placed in the neighborhood of Aranjuez,

Medellín and egg-laying substrates were collected weekly. Eggs were hatched by submerging egg-laying substrates in water, and the species of emerging adults were identified using morphological characteristics. *Aedes aegypti* adults from individual ovitraps were allowed to mate, and DNA extraction of female progeny was performed as follows: individuals were macerated in 50 μL STE (100 mM NaCl, 10 mM Tris-HCl, pH 8.0. 1 mM EDTA) and 1 μL of proteinase K (20 mg/mL; Invitrogen, Waltham, USA) was subsequently added. Samples were incubated at 56 °C for 1 h, followed by 95 °C for 15 min. Isolated DNA was used to confirm the species using *Ae. aegypti*-specific PCR primers[99] (aegF 5′ – CTC TGC GTT GGA TGA ATG AT – 3′; aegR 5′ – ATA GCG TGG TAG CCG TAT G – 3′), and to determine *Wolbachia* infection status using primers specific to the IS5 repeat element[24] (IS5F 5′– GTA TCC AAC AGA TCT AAG C-3′; IS5 5′– ATA ACC CTA CTC ATA GCT AG – 3′). A *Wolbachia*-positive colony was established, and we sequenced a portion of the *wsp* gene using primers reported in ref. [100] (81F – TGG TCC AAT AAG TGA TGA AGA AAC; 691R – AAA AAT TAA ACG CTA CTC CA) and used the Basic Local Alignment Search Tool (BLAST) at https://blast.ncbi.nlm.nih.gov/Blast.cgi to verify that wMel was the infecting *Wolbachia* strain. *Wolbachia*-positive females were backcrossed with Thai strain males for seven generations to produce a *Wolbachia*-infected strain in the Thai genetic background. Given the decline in *Wolbachia* density in eggs during storage[101], we hatched eggs monthly to maintain our colony and tested 30–40 individuals by PCR to ensure infection status prior to our assays.

**Fecundity and fertility assays**. Males and females were mass mated in an 8 L cage in a 1:1 male: female ratio (25 females per cage); although a proportion of females re-mate when mass mated, multiple insemination does not significantly influence fecundity[49] or total sperm stored[68] in Thai strain females. After 24 h, males were removed, and females were blood-fed on the arm of a volunteer. Blood feeding on human subjects was approved by the Comité de Bioética Sede de Investigación Universitaria (Universidad de Antioquia), and volunteers signed a consent form; all methods were performed in accordance with the relevant guidelines and regulations. Four days after blood-feeding, females were individually aspirated into 50 mL conical tubes with a 13 × 4 cm paper towel strip and 6 mL of type II H$_2$O. The strip was removed after 48 h, and eggs were counted using a ZEISS Stemi 508 stereo microscope (ZEISS, Oberkochen, Germany). Eggs were partially dried and stored in an incubator until hatching, which occurred 5–7 days later. To hatch the eggs, the paper strip was placed into a 40 mL cup, filled with type II H$_2$O, supplemented with a pinch of active yeast, and placed under a vacuum for 30 min. The resulting larvae were counted 4–6 days later. Hatch percentage was calculated as larvae/number of eggs; females that laid zero eggs were omitted from the analysis.

**Re-mating assays**. Females were first mated to Thai or Thai$^{Wolb}$ males in parallel and then given the opportunity to re-mate with a DsRed male. We observed the first mating by placing a single male and female into an 8 L cage until a copulation occurred, defined as genitalia engagement for ≥10 s[59,60]. After uncoupling, females were immediately aspirated into a separate 8 L cage with 25 DsRed males until a 1:1 male-female ratio was reached; the second mating opportunity lasted 4 h, after which males were removed. Females were blood-fed and placed into egg-laying chambers 4 days after blood-feeding and given 2 days to lay eggs. After egg-laying, females were frozen at −80 °C until dissections commenced to identify multiply inseminated females; eggs laid by multiply mated females were hatched as previously described. To determine long-term refractoriness, females were mated to a Thai or Thai$^{Wolb}$ male as described above and subsequently placed into a cage with DsRed males 24 h or 7 days later. Identification of multiply mated females was determined by dissection of the lower reproductive tract in 1X PBS to detect the presence (re-mated) or absence (not re-mated) of DsRed sperm[49,60] using a Nikon Eclipse Ti-U fluorescent microscope (Nikon Instruments Inc., Tokyo, Japan).

**Sperm quantification**. To determine if *Wolbachia* infection alters sperm production, we quantified sperm from the seminal vesicles of virgin males, the organs that store mature sperm that are transferred to females during mating[69]. To assess total sperm transfer, we quantified sperm from the bursa (where males deposit the ejaculate[70]) immediately after insemination. To assess the total sperm stored by mated females, we quantified sperm from the spermathecae, the long-term sperm storage organs[68,70], 24 h after mating. Matings to determine sperm transfer were observed to ensure females only copulated once, while matings to determine sperm quantity in the spermathecae were performed as previously described. Our assays utilized adults from the same hatch, and matings were performed on the same day. To quantify sperm transfer, females were flash-frozen on dry ice immediately after uncoupling. To quantify spermathecal sperm quantity, females were mated and placed in the incubator for 24 h. Adults were stored at −80 °C until dissections commenced. Sperm were isolated using a modified protocol reported in ref. [102]. Briefly, tissues were dissected in 1X PBS, placed into a 250 μL chamber containing 100 μL of 1X PBS, ruptured with minutiae pins to release sperm and mixed by pipetting up and down. An additional 100 μL of PBS was added and the solution re-mixed. Ten 5 μL aliquots were placed onto a glass slide and dried for 5 min at 50 °C. Sperm were fixed in 70% ethanol, and sperm nuclei were stained with Giemsa dye (Merck, Kenilworth, USA). Sperm in each drop were counted under

brightfield illumination at 200X magnification. This subsample was used to calculate total sperm[68].

**Longevity**. Females were mass mated in a 1:1 ratio, as previously described, and males were removed prior to the start of our assays. Virgin and mated females were placed into separate 8 L cages (~50 individuals per cage) and kept in the incubator for the duration of the experiment. Two biological replicates were performed for each mating combination or virgin female assessed using individuals from independently hatched cohorts. The sugar solution was replaced weekly. Dead individuals were removed every 3 days until all individuals had perished.

**Labeling *Aedes aegypti* with $^{15}$N**. To examine differences in SFP quantities transferred to females by Thai and Thai$^{Wolb}$ males, we labeled females with the natural isotope heavy nitrogen[71,72] ($^{15}$N) and identified male-derived proteins transferred at mating as in ref. [72]. Briefly, baker's yeast (*Saccharomyces cerevisiae*; LEVAPAN, Sabaneta, Colombia) was inoculated in 200 mL of minimal medium— 20 g/L D-glucose, 1.7 g/L yeast nitrogenous base without amino acids and 5 g/L ammonium sulfate with $^{15}$N (Cambridge Isotope Laboratories, Andover, USA) in sterile water—and incubated on a shaker at 190 rpm and 30 °C for 24 h, after which an additional 800 mL of minimal media was added. Yeast was incubated until a density of 10$^9$ cells/mL was reached. Yeast was harvested by centrifugation at 8000 rpm at 4 °C for 10 min. The pellet was washed with type II H$_2$O, type I H$_2$O, and finally with 1X PBS. Yeast was resuspended in 20 mL of 1X PBS to generate a slurry to feed the mosquito larvae[72]. The slurry was stored at 4 °C and used shortly after its preparation.

Thai strain eggs were hatched as previously described, reared at a density of 200/L, and fed with 1 mL of yeast slurry each day for 5 days. As previous experiments with $^{15}$N-labeled *Ae. aegypti* showed that the first cohort produced adults incapable of flight[71], larvae were grown in 200 mL of rearing water from a previous cohort and 800 mL type II H$_2$O[71,72]. Pupae were transferred to 5 mL tubes, and resulting adults were separated by sex into 8 L cages upon eclosion.

**Ejaculate collection**. $^{15}$N-labeled females were mated to unlabeled, normally reared Thai$^{Wolb}$ or Thai males. A single male and female were placed in an 8 L cage until a copulation occurred. Mated females were flash-frozen on dry ice immediately after uncoupling and stored at −80 °C until dissections commenced. Bursae from experimental (mated to Thai$^{Wolb}$ males) and control females (mated to Thai males) were dissected in 1X PBS with protease inhibitors (cOmplete Mini Protease Inhibitor Cocktail; Roche, Basel, Switzerland). Twenty bursae were collected from control and experimental females (three biological replicates each). After dissection, an equal volume of 2X Laemmli buffer + 5% β-mercaptoethanol was added to each sample. Proteins were solubilized by sonicating with an Elmasonic S30H sonicator (Elam, Singhem, Germany) for 30 s, heated to 95 °C for 15 min, and sonicated again for 30 s. Samples were centrifuged for 10 min at 10,000 × g at 4 °C, and the supernatant was placed into a fresh tube and stored at −80 °C.

**Tandem mass spectrometry proteomic characterization**. Solubilized samples were separated by 1D SDS-PAGE to generate six fractions per sample, digested with trypsin, and analyzed by LC-MS/MS on an Orbitrap Lumos mass spectrometer. Peptides were loaded onto a PepMap 100 C18 pre-column (5 μm particle, 100 Å pore, 300 μm × 5 mm, Thermo Scientific) at 10 μL/min for 3 min with 0.1% formic acid. Peptides were separated on a reverse-phase nano EASY-spray C18 analytical column (2 μm particle, 100 Å pore, 75 μm × 500 mm; Thermo Fisher Scientific, Waltham, USA) with a gradient of 1.6 to 32% acetonitrile in 0.1% formic acid over 120 min at a flow rate of 300 nL/min. All *m/z* values of eluting ions (range 375–1500 Da) were measured at a resolution of 120,000. The MS1 scan was followed by data-dependent MS2 scans (3 s cycle time) to isolate and fragment the most abundant precursor ions at 35% NCE. Fragment ions were measured at a resolution of 15,000. Ions with +1 or unassigned charge were excluded from the analysis, and the dynamic exclusion of previously interrogated precursor ions was 70 s.

Raw spectral data were searched against the *Ae. aegypti* protein database (assembly AaegL5.0), appended with the cRAP v1.0 contaminant database (thegpm.org), using the standard workflow in PEAKS X+ (de novo + PEAKS DB; Bioinformatics Solutions Inc.). Spectral data from all six samples (three control with Thai males and three with Thai$^{Wolb}$ males) were run together in a combined analysis using the following search parameters: mass tolerance of 15 ppm for parent ions and 0.5 Da for fragment ions, carbamidomethylation (C) as a fixed modification, oxidation (M) and deamidation (NQ) as variable modifications, and up to three missed tryptic cleavages. Peptide identifications were filtered to a false discovery rate (FDR) < 1% based on the decoy-fusion approach[103]. Protein identifications were filtered to a −10lgP score ≥ 20 and at least one unique peptide-spectrum matches (PSMs). Label-free quantitative comparisons were based on the relative abundance of peptide features using the PEAKS Q module (Bioinformatics Solutions Inc., Waterloo, Canada). Additionally, raw spectra from the three samples from matings with Thai$^{Wolb}$ males were searched directly against *Wolbachia pipientis* wMel (UP000008215), resulting in 45,120 PSMs and 125 proteins (Supplementary Data 1).

**Statistics and reproducibility**. R statistical software version 3.6.1 coupled with R-Studio Version 1.2.1335 was used for all analyses[104]. The number of eggs laid by each female (fecundity), seminal vesicle and spermathecal sperm quantity data were first analyzed to determine the probability distribution that fit the data, including normal, negative binomial, and Poisson distributions. The Akaike information criterion (AIC) was used to compare the best distribution that fit the data, where the lowest AIC value corresponds to the best-fitted distribution. We also evaluated if each replicate of our assays differed in the characteristics we were examining. Because no significant differences were found between replicates, data from each experiment were combined and replicate used as a random factor in the models. We analyzed fecundity, seminal vesicle, bursa and spermathecal sperm quantity using a linear mixed model (LMM) using the mating combination as the fixed factor. For hatch percentage, a generalized linear mixed model (GLMM) with a binomial distribution was used with mating combination as a fixed factor.

Re-mating incidence was evaluated by performing the chi-square test of independence based on the contingency table of two variables—*Wolbachia* infection status and mating status (re-mated and not re-mated)—using the R statistical package and chi q test function. Adult longevity was analyzed using a Kaplan–Meier curve to illustrate the cumulative survival probability over time. Cox proportional hazards (PH) regressions and log-rank tests were used to evaluate differences between mating combinations. Wing sizes were analyzed using a LMM with wing size as the response variable, mosquito strain as a fixed factor and replicate as a random factor in the model. R code supporting this manuscript will be made available upon request.

**Reporting summary**. Further information on research design is available in the Nature Portfolio Reporting Summary linked to this article.

## Data availability

Data to assess fertility, post-mating responses, and sperm transfer/storage are available in Supplementary Data 2. The mass spectrometry proteomics data have been deposited to the ProteomeXchange Consortium (http://proteomecentral.proteomexchange.org) via the PRIDE partner repository[105] with project accession PXD043965. RNA sequencing data from Degner et al.[72] can be accessed at the Sequence Read Archive (SRA), accession number SRP158536. All other data are available from the corresponding author upon reasonable request.

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

## Acknowledgements

We thank members of the Avila lab for their help in mating assays and data collection, Sebastián Diaz for help in establishing our original *Wolbachia* positive colony, David Andres Borrego Muñoz for preliminary statistical analysis, Laura Harrington and Omar Triana for providing the mosquito strains used in our experiments, Margarita Correa and Giovan F. Gómez for technical advice, Seth Bordenstein, Sarah Bordenstein and Rupinder Kaur for helpful discussion, Mike Deery and Renata Feret of the Cambridge Center for Proteomics (https://proteomics.bio.cam.ac.uk/) for assistance in sample preparation and quality control, and Ruta N Medellín for laboratory support. This work was supported by the COLCIENCIAS, Universidad de Antioquia and Max Planck Society cooperation grant 566-1-2014 (to F.W.A.), Minciencias grant 821-2019 (to C.A.P. and F.W.A.), and National Science Foundation grant DEB 1655840 (to S.D.).

## Author contributions

J.O., F.W.A., C.A.P. and S.D. conceived and designed the experiments; G.R.U. collected *Wolbachia*-infected individuals from the field; C.B. performed the genetic backcross; J.O. performed the fecundity, fertility, longevity, and receptivity assays; J.O., F.W.A., L.F.R.S. and L.B. performed the sperm quantification assays; S.V.A. labeled female *Ae. aegypti* with $^{15}$N; F.W.A., C.A.P. and S.V.A. mated $^{15}$N-labeled females with unlabeled males, performed the tissue dissections, and prepared protein extracts for mass spectrometry; J.O., C.C. and S.D. analyzed the data; J.O. and C.C. prepared the figures, F.W.A. wrote the manuscript, and all authors reviewed and approved the manuscript.

## Competing interests

F.W.A. is an Editorial Board Member for *Communications Biology* but was not involved in the editorial review of, nor the decision to publish this article. All other authors declare no competing interests.
