## [Peer Review File · Communications Biology]

Reviewers' comments:

Reviewer #1 (Remarks to the Author):

A central tenant of this work is that the success of methods designed to control *Ae. aegypti* populations is largely dependent on minimal effects on the fitness of mosquitoes released for the purpose of control. The authors focus on a method that relies on infecting mosquitoes with *wolbachia*, which induces cytoplasmic incompatibility in hybrids in matings of infected released individuals with those found in the natural populations. Importantly, infected females display refractoriness to medically relevant arboviruses. However, the effectiveness of the method is dependent on the reproductive fitness of the infected mosquitoes, and a thorough exploration of postmating effects arisen from mating to an infected partner is much needed. Given the overwhelming amount of information on the importance of postmating responses in insects, the work tackles an important problem. Using a series of well-designed experiments, the authors find that infection affects female fecundity, fertility, and re-mating incidence. A proteomics analysis finds that only a handful *Wolbachia* proteins are transferred to females by infected males.

I have only a few comments.

While the emphasis is on the effectiveness *wolbachia* infections as a method of control (see for example end of the introduction), I think the work provides some very interesting findings in terms of comparative biology to other insects. I believe those findings could be of particular interest to the wider audience that reads *Communication Biology* articles. The authors touch upon that, here and there, but a better highlight and even the inclusion of a table that compares, at least *Drosophila*, to the findings of this study would be a welcomed addition. This is important to strengthen the relevance for people in the discipline and for a broader audience.

Your findings about remating being affected (Table 1) are very interesting but unfortunately several crosses were not statistically significant. I agree with you that there is a clear trend of an increase in remating incidence when females first mate to infected males. However, someone can argue that your results were non-significant. I noticed that if I pool all the data together the result is significant. Do you have proportion of remating over time? You say all remating happens within two hours, so I was wondering whether there is also a time effect. Moreover, there is something that might be of interest to control programs, and that you could speculate in the discussion. Your significant result of increased remating happens when the females are of the same population origin as infected males. If the infected males are rare to the females (e.g. Thai males with Rockefeller females), you get less of an effect (and non-significant). Not sure about mosquitoes, but there are reports of females preferring rare males. Ideally, the comparison should have included uninfected Rockefeller males, nevertheless you could speculate that a venue worth pursuing might be whether control programs would benefit from releasing infected males from foreign populations to minimize the unwanted increase remating that might otherwise take place. It leaves me wondering what might have happened to fertility and fecundity if you would have tried females from different populations than infected males.

You cite two papers to claim that previous work have found modest effect of infection on male and female fertility (abstract and line 86). Later you state that "Previous studies have noted a suppression in fertility" (line 294). You need to better clarify how your findings are different from previous ones. I quickly looked at the previous work and my interpretation is that both studies claim that infections reduce the fecundity of infected females and viability of eggs. If that is the case (more in line with what you state in line 294), then your previous claims are an overstatement. I do not think that that (i.e. that previous studies have also found, as you do here, an effect on fertility and fecundity) deters from the importance and relevance of looking at other PMRs in infected mosquitoes. Moreover, your work appears to do a better job in including both infected and uninfected partners, and also offers a nice survey of proteome changes.

The proteomics analysis is a novel addition that provides a list of interesting candidate proteins for future studies. Supplementary data set1 lists the 125 proteins that were present in the infected male ejaculates. The data shows GO terms. Did you conduct a formal analysis of GO-enrichment? You state the proteins fell into different classes, but a formal analysis might highlight enriched ones.

Reviewer #2 (Remarks to the Author):

General comments -

Wolbachia bacteria live inside the tissues of insects where they can have wide-ranging effects on insect reproduction. *Aedes aegypti* transinfected with the wMel strain of Wolbachia have been introduced to local mosquito genetic backgrounds and mass-released for release programs around the world. These releases have greatly reduced arbovirus transmission because Wolbachia can block virus transmission and spread through populations via cytoplasmic incompatibility and maternal transmission. The phenotypic effects of Wolbachia strains have been investigated in detail by a number of studies, with effects ranging from reductions in fertility, longevity and costs to larval development, to effects on blood feeding behaviors and stress tolerance. These effects are very much dependent on the mosquito genotype and environmental conditions.

In this study, the authors introduce the wMel strain from a field-collected population to a Thai *Ae. aegypti* genotypes through backcrossing, then investigate effects of Wolbachia on post-mating responses including fertility, remating, longevity and sperm transfer/storage. They find moderate costs of wMel infection to female fertility, particularly in terms of egg hatch. They also find that female mosquitoes are more likely to remate if they first mate with Wolbachia-infected males despite no clear effect of Wolbachia infection on sperm transfer and storage. They then investigate the seminal proteome of Wolbachia-infected and uninfected mosquitoes and identify paternally transmitted Wolbachia proteins. While previous studies have tested for effects of Wolbachia on mosquito fertility in different mating combinations (see Turley et al. 2013 *Parasites and Vectors*, Dutra et al. 2015 *PLoS NTDs*) and effects of Wolbachia on sperm quantity and viability (Turley et al. 2013), the current study provides some interesting insights into Wolbachia effects on mosquito remating. While a relatively small effect, the fact that mosquitoes are more likely to remate after mating with Wolbachia-infected males could have significant implications for the long-term effectiveness of Wolbachia release programs (e.g. if wild females can restore their compatibility after mating with an incompatible male). However, remating on its own is not very informative without looking at the fertility of these remated females. It would have been worthwhile for the authors to test whether females that remate after mating with an incompatible male first are able to produce viable offspring. If not, then mosquitoes remating won't be an issue for Wolbachia release programs since CI will still occur based on the first mating.

The authors also find some interesting effects of Wolbachia infection on adult longevity, though I am concerned here about the level of replication. From what I can tell from the methods, only a single cage (with a group of 50 mosquitoes) was used for each treatment so the replicate mosquitoes are not being reared independently. Without replication at the cage level, there could be confounding effects of cage that contribute to mosquito lifespan.

Minor comments -

Figure 1 – A couple of data points appear to be above 1.0 – please check the raw data for calculation errors since hatch proportions cannot be above 1.

Figure 2 – The x axis has been slightly cropped so that the x-axis title is partially obscured

Line 307 – Note that fitness costs and benefits of Wolbachia often differ between studies, which could be due to different rearing conditions or the use of different mosquito lines.

Line 366 – 372 – This section is introductory and is not appropriate for a conclusions paragraph- I suggest deleting since it is covered elsewhere.

Line 475 - Longevity experiment – please clarify about the level of replication- was it a single cage per treatment?

Reviewer #3 (Remarks to the Author):

Through insect studies the authors find Wolbachia infections can decrease fecundity through reduced egg laying and hatching, increase female re-mating, and change longevity trends of the females in relation to mating status. They followed this with an investigation into sperm production, transfer and female storage where no differences were found. Finally, a proteomic approach illustrated a decrease in SFP trypsin-7 in the sperm of Wolbachia-infected males and an absence of paternally transferred cytoplasmic incompatibility factor proteins (Cifs).

I found this paper to be written in a clear, concise manner with simple, informative figures. There is a nice mix of insect studies with proteomic work to support their ideas. It highlights what information is missing in the literature, how this paper fits into the field and, future projects that could strengthen the mechanistic understanding of Wolbachia infections in *Ae. aegypti*. My comments are for small changes that would help to clarify the authors' choices and for ease of understanding for the reader.

The authors have provided insight into previously unknown effects of a Wolbachia infection in *Aedes aegypti* females. This is key information for the planning, running and success of long-term control programs at a time when insecticide resistance has hampered other control options.

Minor comments

1. Could you explain how the Thai *Aedes aegypti* strain compares to wild-type mosquitoes or those being used in control programmes? This would help to understand how the findings of this paper can be applied to the strains of *Ae. aegypti* mosquitoes in the field. Alternatively, this could also be an explanation of why the Thai strain was chosen for this work.
2. Line 116: Could you explain the acronym of Cif protein as this is the first mention in the body of the paper.
3. Line 166: Details on the other mosquito stain backgrounds would be nice. For example, the insecticide resistance/susceptibility status of the Acacias and Rockefeller strains.
4. Line 152: Figure 1: To make this easier for the reader, colour coding could be used for the different mating combinations.

5. Line 216: Figure 3: To make this easier for the reader, colour coding could be used for the different mating combinations.

6. Line 479-480: A small addition to justify why sugars were replaced weekly and dead mosquitoes removed from the study arena every 3 days (rather than more frequently).

Reviewer #1:

While the emphasis is on the effectiveness of *Wolbachia* infections as a method of control (see for example end of the introduction), I think the work provides some very interesting findings in terms of comparative biology to other insects. I believe those findings could be of particular interest to the wider audience that reads *Communication Biology* articles. The authors touch upon that, here and there, but a better highlight and even the inclusion of a table that compares, at least *Drosophila*, to the findings of this study would be a welcomed addition. This is important to strengthen the relevance for people in the discipline and for a broader audience.

We have included a new Table to our Discussion that compares the PMRs we assessed in our study to those reported in *Drosophila melanogaster* as well as effects on PMRs reported in other transinfected insects. Given that various *Wolbachia* strains have been examined in, kept our focus on how infection by the wMel strain of *Wolbachia* alters female PMRs in these species (Table 3; Line 333).

Your findings about remating being affected (Table 1) are very interesting but unfortunately several crosses were not statistically significant. I agree with you that there is a clear trend of an increase in remating incidence when females first mate to infected males. However, someone can argue that your results were non-significant. I noticed that if I pool all the data together the result is significant. Do you have proportion of remating over time? You say all remating happens within two hours, so I was wondering whether there is also a time effect.

We did not intend to suggest that all re-mating occurs within the first 2 h of an initial mating. Re-mating incidence by Thai strain females is highest (~25%) when females are exposed to competitor males in the first 2 h post-mating. Re-mating incidence declines when exposure occurs later; around 10% of females' re-mate when exposed to males 4-12 h after the initial mating but falls to nearly 0% when exposure occurs 20 h and does not occur when exposure occurs after the first day (10.4269/ajtmh.15-0893).

Female re-mating is likely to occur shortly after an initial mating, as female refractoriness is observed as quickly as 5 min in Orlando strain *Ae. aegypti* (10.1016/j.cub.2017.10.074). Thus, there is an effect of time in mating inhibition induced by SFP receipt, but we did not test this here. However, we did determine that mating inhibition is 100% by 24 h and lasts for up to 7 days post-mating showing that although Thai^{Wolb} males are suboptimal at preventing re-mating in their mates in the short-term, they can inhibit mating completely by 24 h and that this effect is long lasting. We have added this information to the manuscript as Table S1. Finally, our re-mating assays might be underestimating re-mating incidence. A recent study that assigned parentage of progeny from mass mated *Ae. aegypti* using microsatellites found that *Ae. aegypti* females' mate with 3-4 partners on average (10.1093/jme/tjac081); we have added this reference to our Discussion (Lines 370-372).

Moreover, there is something that might be of interest to control programs, and that you could speculate in the discussion. Your significant result of increased remating happens when the females are of the same population origin as infected males. If the infected males are rare to the females (e.g. Thai males with Rockefeller females), you get less of an effect (and non-

significant). Not sure about mosquitoes, but there are reports of females preferring rare males. Ideally, the comparison should have included uninfected Rockefeller males, nevertheless you could speculate that a venue worth pursuing might be whether control programs would benefit from releasing infected males from foreign populations to minimize the unwanted increase remating that might otherwise take place. It leaves me wondering what might have happened to fertility and fecundity if you would have tried females from different populations than infected males.

This is an interesting point, but one we would rather not mention since control programs (either population replacement or population suppression) introduce the *Wolbachia* strain being used into a local *Aedes aegypti* strain to better insure the survival of the released adults—It has been shown that matching the genetic background of the liberated adults enhances their survival as local populations can display resistance to commonly used insecticides in the areas where releases occur (10.1371/journal.pntd.0007023).

You cite two papers to claim that previous work have found modest effect of infection on male and female fertility (abstract and line 86). Later you state that “Previous studies have noted a suppression in fertility” (line 294). You need to better clarify how your findings are different from previous ones. I quickly looked at the previous work and my interpretation is that both studies claim that infections reduce the fecundity of infected females and viability of eggs. If that is the case (more in line with what you state in line 294), then your previous claims are an overstatement. I do not think that that (i.e. that previous studies have also found, as you do here, an effect on fertility and fecundity) deters from the importance and relevance of looking at other PMRs in infected mosquitoes. Moreover, your work appears to do a better job in including both infected and uninfected partners, and also offers a nice survey of proteome changes.

The Reviewer is correct that we have contradicted ourselves given the results you point out. To better state our results, and given the study brought to our attention by Reviewer #2 (see below), we have modified our language to state clearly that previous studies have shown a suppression in fertility when both sexes are infected, as well as when only a single sex is infected (Lines 321-323).

The proteomics analysis is a novel addition that provides a list of interesting candidate proteins for future studies. Supplementary data set1 lists the 125 proteins that were present in the infected male ejaculates. The data shows GO terms. Did you conduct a formal analysis of GO-enrichment? You state the proteins fell into different classes, but a formal analysis might highlight enriched ones.

We performed a GO analysis on the paternally transferred *Wolbachia* proteins we identified and found that only 4 GO terms were enriched. This has been added to the Results section (Lines 287-289) and we added a Supplementary Figure (Figure S4).

Reviewer #2:

While previous studies have tested for effects of *Wolbachia* on mosquito fertility in different

mating combinations (see Turley et al. 2013 Parasites and Vectors, Dutra et al. 2015 PLoS NTDs) and effects of Wolbachia on sperm quantity and viability (Turley et al. 2013), the current study provides some interesting insights into Wolbachia effects on mosquito remating. While a relatively small effect, the fact that mosquitoes are more likely to remate after mating with Wolbachia-infected males could have significant implications for the long-term effectiveness of Wolbachia release programs (e.g. if wild females can restore their compatibility after mating with an incompatible male). However, remating on its own is not very informative without looking at the fertility of these remated females. It would have been worthwhile for the authors to test whether females that remate after mating with an incompatible male first are able to produce viable offspring. If not, then mosquitoes remating won't be an issue for Wolbachia release programs since CI will still occur based on the first mating.

This is a valid point and one that we examined but did not include in our initial submission. To ensure that multiply mated females initially mated to an incompatible male generated viable progeny, we hatched the eggs from multiply mated Thai, Acacias and Rockefeller females from our re-mating assays. We found that although these females had suppressed fertility, they produced viable progeny when they utilized sperm from the second, uninfected male. This result aligns with our previous work that shows that multiply mated females produce mixed progeny, but primarily use sperm from the first mating male (10.3389/fphys.2021.691221). We have included this data in the Results (Lines 181-188) and added a Supplemental Figure (Figure S2).

The authors also find some interesting effects of Wolbachia infection on adult longevity, though I am concerned here about the level of replication. From what I can tell from the methods, only a single cage (with a group of 50 mosquitoes) was used for each treatment so the replicate mosquitoes are not being reared independently. Without replication at the cage level, there could be confounding effects of cage that contribute to mosquito lifespan.

In Figure 2A, we performed two replicates of our assays with adults from independently hatched cohorts. The data presented in Figure 2B was a last-minute addition, and due to a miscommunication between me and the lead author, was data from only a single replicate. We have removed this portion of the figure and replaced it with an assay we performed to assess if the longevity effects *Wolbachia* on virgin females were dependent on adult nutrition; we gave females access to the standard adult nutrition (10% sucrose) or poor adult nutrition (2% sucrose). We performed two replicates of each treatment using females from independently hatched cohorts as previously (giving us 4 replicates that show that Thai^{Wolb} virgin females are longer lived than Thai virgin females under standard conditions with 10% sucrose). On both diets, Thai^{Wolb} females lived significantly longer than uninfected females (Figure 2B). This data has been added to the manuscript (Lines 200-206).

Minor comments -

Figure 1 – A couple of data points appear to be above 1.0 – please check the raw data for calculation errors since hatch proportions cannot be above 1.

This sometimes happens due to error when the number of eggs counted is less than the number of larvae counted after hatching the eggs (normally a small difference of 1 or 2 individuals). We

looked at the raw data and found that for one female in one replicate the number of larvae counted exceeded the number of eggs by 2. Because this type of error should affect a small number of females but should be consistent (but only noted when Hatch % exceeds 100%), we would prefer to keep the figure as is to maintain the same error for all females.

Figure 2 – The x axis has been slightly cropped so that the x-axis title is partially obscured.

We have corrected this.

Line 307 – Note that fitness costs and benefits of *Wolbachia* often differ between studies, which could be due to different rearing conditions or the use of different mosquito lines.

We have done this (Lines 321-323)

Line 366 – 372 – This section is introductory and is not appropriate for a conclusions paragraph- I suggest deleting since it is covered elsewhere.

We have done this.

Line 475 - Longevity experiment – please clarify about the level of replication- was it a single cage per treatment?

Please see our answer to this above.

Reviewer #3:

1. Could you explain how the Thai *Aedes aegypti* strain compares to wild-type mosquitoes or those being used in control programmes? This would help to understand how the findings of this paper can be applied to the strains of *Ae. aegypti* mosquitoes in the field. Alternatively, this could also be an explanation of why the Thai strain was chosen for this work.

Control programs that release *Wolbachia*-infected individuals typically place *Wolbachia* into a local strain prior to their release (e.g., [10.1371/journal.pntd.0007023](https://doi.org/10.1371/journal.pntd.0007023)). Thus, population control programs use a variety of different strains, making this a difficult question to answer. However, in a previous study we used the Thai strain and the Acacias strain when the latter was close to the field (F7) and found that fecundity, fertility, and ejaculate depletion were similar between strains ([10.1016/j.jinsphys.2020.104019](https://doi.org/10.1016/j.jinsphys.2020.104019)).

Our primary objective in using Thai strain is that numerous studies that have examined *Ae. aegypti* fertility and female PMRs have used this strain. This includes the identification of the sperm and seminal fluid proteomes, and the testis and male accessory gland (MAG) transcriptomes ([10.1074/mcp.RA118.001067](https://doi.org/10.1074/mcp.RA118.001067)), characterization of MAG SFP synthesis ([10.1016/j.jinsphys.2014.07.004](https://doi.org/10.1016/j.jinsphys.2014.07.004)), identification of mating-induced gene expression changes in the lower female reproductive tract ([10.1371/journal.pntd.0004451](https://doi.org/10.1371/journal.pntd.0004451)) and spermathecae

(10.1038/s41598-020-71904-z), gene expression changes in the lower female reproductive tract that result from SFP receipt (10.1186/s12864-021-08201-0), quantification of sperm transfer during mating and subsequent female sperm storage (10.3389/fitd.2022.816556), the roles of SFPs in oviposition and female longevity (10.1016/j.jinsphys.2018.05.001) and the time dependent effects of female re-mating incidence (10.4269/ajtmh.15-0893). Given the quantity of information known regarding reproduction in this strain, our hope is that we can better ascertain why alterations to the female PMR are occurring in *Wolbachia*-infected females.

2. Line 116: Could you explain the acronym of Cif protein as this is the first mention in the body of the paper.

We have done this.

3. Line 166: Details on the other mosquito stain backgrounds would be nice. For example, the insecticide resistance/susceptibility status of the Acacias and Rockefeller strains.

We have included a line in the methods (Lines 442-443) that states that the Acacias strain is resistant to pyrethroid insecticides while the Rockefeller strain is susceptible and have cited the study that tests the susceptibility of these strains (10.1371/journal.pntd.0010001); we obtained the Acacias and Rockefeller mosquitoes used in our study from the lab that reported this study.

4. Line 152: Figure 1: To make this easier for the reader, colour coding could be used for the different mating combinations.

We have used color coding for the different mating combinations.

5. Line 216: Figure 3: To make this easier for the reader, colour coding could be used for the different mating combinations.

We have used color coding for the different mating combinations.

6. Line 479-480: A small addition to justify why sugars were replaced weekly and dead mosquitoes removed from the study arena every 3 days (rather than more frequently).

We commonly replace sugar solution weekly in our assays that persist for long durations which has worked well for us (e.g., when assessing long-term refractoriness, or the effects of male (10.3389/fphys.2021.691221) or female age (In preparation) on mosquito fertility). Examining dead individuals every 3 days was a decision made by the lead author initially and kept to maintain cohesion among our replicates.

(Reviewer #1)

I am satisfied with how the authors have revised the original submission and addressed my questions and comments in their response.

(Reviewer #2)

The authors have addressed all my comments and I have no further suggestions